# Imaging of Fibrous Dysplasia Protuberans, an Extremely Rare Exophytic Variant of Fibrous Dysplasia

**DOI:** 10.3390/diagnostics11061006

**Published:** 2021-05-31

**Authors:** Amin Haghighat Jahromi, William F. James, Michael D. Starsiak, Eugene D. Silverman

**Affiliations:** 1Department of Radiology/Nuclear Medicine, UCSD Medical Center, 200 W Arbor Drive, San Diego, CA 92103, USA; 2Department of Radiology, Nuclear Medicine Division, Naval Medical Center San Diego, 34800 Bob Wilson Drive, San Diego, CA 92134, USA; william.f.james26.mil@mail.mil (W.F.J.); michael.d.starsiak.mil@mail.mil (M.D.S.); eugene.d.silverman.civ@mail.mil (E.D.S.)

**Keywords:** fibrous dysplasia, SPECT/CT, exophytic bone lesion

## Abstract

This paper details the case report of a 26-year-old man who presented with a growing right-sided skull mass evaluated with ultrasound, non-contrast CT, contrast-enhanced MRI and ^99m^Tc-MDP whole body bone scan with SPECT/CT. These studies suggested a broad differential diagnosis favoring benign osseous lesions. Given a more recent increase in the rate of growth, headache and large size, the lesion was excised via craniotomy followed by cranioplasty. Pathology confirmed fibrous dysplasia (FD) as the diagnosis. Interestingly, this report is the imaging evaluation of the exophytic subtype of FD, the so-called FD protuberance, an extremely rare variant of FD, of which only two case reports are found in the literature.

**Figure 1 diagnostics-11-01006-f001:**
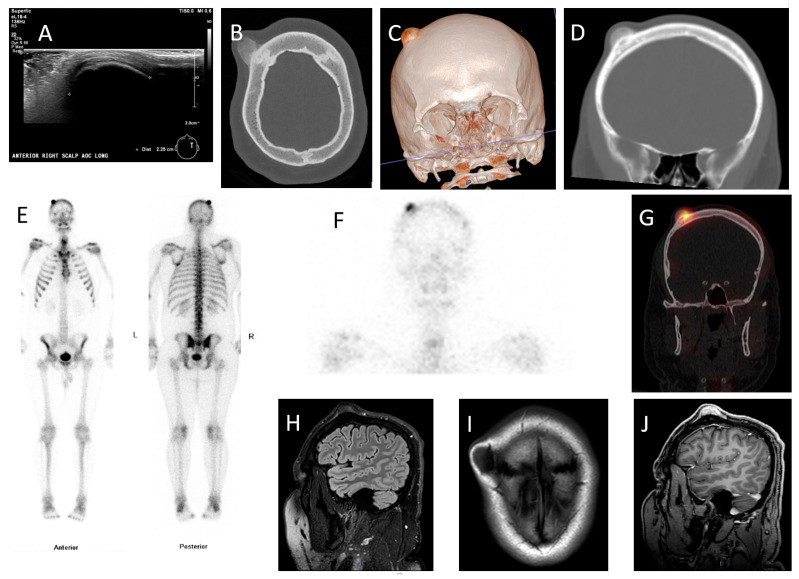
(**A**) 26-year-old man presented with a right-sided skull mass that had steadily increased in size since he first noticed it 5 years ago. After a 2–3-fold increase in the rate of growth over the prior year, he began to complain of bi-temporal headaches. (**A**) Ultrasound imaging revealed a 2.3 cm subcutaneous anechoic lesion with strong posterior acoustic shadowing and without vascularity. (**B**–**D**) Axial CT slice, 3D volume rendered CT and coronal CT slice revealed an exophytic lesion with ground-glass matrix on the surface of the outer table of the right coronal suture. (**E**) ^99m^Tc-MDP whole body bone scan revealed intense focal uptake on the right side of the skull. (**F**) Coronal reconstruction of bone scan SPECT images localized the focal uptake anteriorly on the right. (**G**) Coronal SPECT/CT images co-localized the intense focal uptake to the exophytic lesion on CT. (**H**,**I**) Representative sagittal FLAIR sequence and axial T1-weighted image revealed a T1 and T2 hypointense exophytic lesion arising from the outer table of the right coronal suture without bone marrow edema. (**J**) Sagittal 3D contrast-enhanced MRI revealed intense homogenous enhancement in the lesion without surrounding invasion. The overall differential diagnosis was broad, favoring benign osseous lesions such as osteochondroma, enchondroma protuberans, and surface osteoma, over less likely malignant lesions such as parosteal osteosarcoma [1]. Given the recent increase in the rate of growth, headache and large size, the lesion was excised via craniotomy followed by cranioplasty. (**K**) is a low-power image showing thin, irregular, curvilinear trabeculae of woven bone, whereas (**L**) is a high-power image showing bland fibroblastic cells and a lack of conspicuous osteoblastic rimming, consistent with fibrous dysplasia (FD). FD is a typically benign osseous lesion, constituting 5% of all benign bone lesions [2]. It is a non-inherited anomaly typically presenting before 30 years of age and involving the craniofacial bones, femur, tibia, ribs and pelvis [2]. In FD, normal bone marrow is replaced by fibro-osseous tissue; therefore, it is typically intramedullary [3]. The exophytic subtype of FD or FD protuberance is an extremely rare variant of FD, with only two case reports in the literature, first described in 1994 [4,5]. This rare variant mimics benign lesions such as osteochondroma, exostosis, surface osteoma, enchondroma protuberance, and even malignant bone tumors such as osteosarcoma and chondrosarcoma. To our knowledge, the FD protuberans variant is not described in the nuclear medicine or radiology literature [4,5,6]. As such, this case report appears to be the first imaging evaluation of FD protuberans [7,8].

## Data Availability

Not applicable.

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
