# Peer review of "Imaging of Fibrous Dysplasia Protuberans, an Extremely Rare Exophytic Variant of Fibrous Dysplasia"

_diagnostics, 2021, doi:10.3390/diagnostics11061006_

Round 1
Reviewer 1 Report
An interesting case with intact image study.
But here is no pathologic figure.
The pathologic slide maybe more important then image H and I.
Author Response
Thank you for the constructive feedback. Pathologic image is now added per your suggestion.
Reviewer 2 Report
This is an interesting very rare case of an FD protuberans lesion, which is of interest for readers of Diagnostics. Includes multimodal images by whole body and local CT, MRI, and ultrasound . In the beginning of the figure description the age is missing. Appears to be the first imaging evaluation of FD protuberans.
Author Response
Thank you for you review. I am not sure about your comment that In the beginning of the figure description the age is missing. The figure caption starts with "A 26-year-old man presented with a right-sided skull mass ...". Maybe there is a glitch in the system, not showing the age.